# Non-Invasive Intra-Abdominal Pressure Measurement by Means of Transient Radar Method: In Vitro Validation of a Novel Radar-Based Sensor

**DOI:** 10.3390/s21185999

**Published:** 2021-09-07

**Authors:** Salar Tayebi, Ali Pourkazemi, Manu L.N.G. Malbrain, Johan Stiens

**Affiliations:** 1Department of Electronics and Informatics, Vrije Universiteit Brussel, 1050 Brussels, Belgium; apourkaz@etrovub.be (A.P.); jstiens@etrovub.be (J.S.); 2First Department of Anaesthesiology and Intensive Therapy, Medical University of Lublin, 20-059 Lublin, Poland; manu.malbrain@janpalfijngent.be; 3Medical Department, Medical Direction, AZ Jan Palfijn, 9000 Ghent, Belgium

**Keywords:** intra-abdominal pressure, non-invasive measurement, microwave reflectometry, transient radar method

## Abstract

Intra-abdominal hypertension, defined as an intra-abdominal pressure (IAP) equal to or above 12 mmHg is one of the major risk-factors for increased morbidity (organ failure) and mortality in critically ill patients. Therefore, IAP monitoring is highly recommended in intensive care unit (ICU) patients to predict development of abdominal compartment syndrome and to provide a better care for patients hospitalized in the ICU. The IAP measurement through the bladder is the actual reference standard advocated by the abdominal compartment society; however, this measurement technique is cumbersome, non-continuous, and carries a potential risk for urinary tract infections and urethral injury. Using microwave reflectometry has been proposed as one of the most promising IAP measurement alternatives. In this study, a novel radar-based method known as transient radar method (TRM) has been used to monitor the IAP in an in vitro model with an advanced abdominal wall phantom. In the second part of the study, further regression analyses have been done to calibrate the TRM system and measure the absolute value of IAP. A correlation of –0.97 with a *p*-value of 0.0001 was found between the IAP and the reflection response of the abdominal wall phantom. Additionally, a quadratic relation with a bias of −0.06 mmHg was found between IAP obtained from the TRM technique and the IAP values recorded by a pressure gauge. This study showed a promising future for further developing the TRM technique to use it in clinical monitoring.

## 1. Introduction

According to the consensus definitions of the Abdominal Compartment Society (WSACS, formerly known as the World Society of the Abdominal Compartment Syndrome, www.wsacs.org (accessed on: 8 February 2021)), intra-abdominal pressure (IAP) is defined as the steady-state pressure inside the abdominal cavity [1]. Reviewing previous investigations, the incidence of intra-abdominal hypertension (IAH), defined as an IAP values equal to or above 12 mmHg, is around 50% within the first week of admission in critically ill patients [2,3,4,5]. IAH results in reduced blood return to the heart, which subsequently, results in reduced cardiac output and blood perfusion to distant organs, and ultimately, multiple organ dysfunction and failure. Moreover, late detection of IAH results in a more severe condition known as abdominal compartment syndrome (ACS); that can be fatal when left untreated and characterized by a sustained IAP value above 20 mmHg and new onset organ failure. Since IAH is associated with higher morbidity and mortality, IAP monitoring is highly recommended in ICU patients to anticipate and predict the development of abdominal compartment syndrome and to provide a better care for patients hospitalized in the ICU [2]. 

Although several direct and indirect IAP measurement techniques have been proposed in the literature [6,7,8], IAP monitoring through the bladder with a maximal instillation volume of 25 mL of sterile saline is the actual reference standard advocated by the abdominal compartment society [9]. In this method, IAP should be monitored at end-expiration with the patient in supine position. Additionally, the zero level for the pressure transducer is the mid-axillary line, where it crosses the iliac crest [10]. Since this measurement method reflects the pressure inside the bladder, it is not applicable to patients with hematomas or pelvic masses [7]. In addition, it is a non-continuous measurement that needs to be performed at least every 6 hours, which increases the risk of late IAH detection. Subsequently, finding a novel non-invasive technique that can also be used for patients with pelvic masses could potentially improve the patient care in the future. Using microwave reflectometry has been proposed as one of the most promising IAP measurement methods [8,11].

Microwave reflectometry is a testing method for the electromagnetic characterization of materials. In this technique, electromagnetic waves with a certain frequency (between 300 MHz and 300 GHz) will be sent towards the sample tissue under test. According to the frequency-dependent dielectric property differences of the materials in each layer of a stratified medium, each interface and layer reflects, transmits, and absorbs the electromagnetic waves in a unique way, offering the possibility to objectively characterize each layer of the full structure. Transient radar method (TRM) can be considered as a sub-group of time-domain microwave reflectometry methodology which deals with the transient part of the reflection signal from the sample under test rather than using its steady-state form [12]. Since IAP elevation influences the mechanical, geometric, and electromagnetic properties of the abdominal wall, this radar-based technique is able to track IAP fluctuations and changes as it is able to extract electromagnetic and geometric properties of the sample under investigation without any prior knowledge [13].

The aim of this study was to introduce the TRM technique and to investigate its robustness in monitoring IAP in an in vitro model. In the first part of this experimental investigation, the different IAP levels within an abdominal wall phantom are monitored by means of TRM and compared to the pressure obtained with a pressure gauge. In the second part, further regression analyses will be done in order to calibrate the TRM output as a function of IAP values obtained in the first part. Subsequently, the absolute value of the IAP within the abdominal wall phantom can be measured via TRM.

## 2. Materials and Methods

### 2.1. Abdominal Wall Phantom

An abdominal wall phantom, manufactured by “The Chamberlain Group” (www.thecgroup.com (accessed on: 7 November 2020), Great Barrington, MA, USA) was used for this investigation, in order to simulate the real abdominal wall structure. The provided phantom consisted of an artificial abdominal compartment in addition to an artificial abdominal wall. The abdominal wall is made of different layers to simulate all the tissues in the human abdominal wall including skin, fat tissue, abdominal muscles, fascia, and peritoneum (Figure 1a). The abdominal wall phantom was developed further; an inflatable air-filled balloon as well as a water-filled balloon were added to the structure to make it more realistic (Figure 1b). Ultimately, IAP could be adjusted using a manual air pump connected to the air-filled balloon. 

### 2.2. Transient Radar Method Set-Up

The TRM measurement set-up is illustrated in Figure 2. This system consisted of a single frequency generator, power divider, reflective Single Pole Single Throw (SPST) switch, amplifier, transmitter and receiver antennas, single shot samplers, low-pass filters, delay creator, and the trigger module. The single frequency generator was a narrow-band voltage-controlled oscillator (VCO) that generates continuous electromagnetic waves with the frequency of 10 GHz. The power divider is the next module that functions as a power splitter that provides a similar (power) amplitude and phase to each port. The SPST switch has two functions in this set-up; the rise time generation which is carried by the generated single harmonic from base-band to intermediate band. 

The other task, however, is to reflect signal for triggering single shot sampler at the toggling moment from conductive to non-conductive condition. The function of the amplifier is to increase the amplitude of signals which are radiated towards the sample under test. Transmitter and receiver antennas are the components that work as the emitter and collector of the electromagnetic waves. The single-shot sampler, also called the sample and hold module, is the module that records the amplitude of the reflected electromagnetic waves during an infinitesimal time interval. The low pass filter has the function to filter out the noise and to increase the throughput of the recording signal procedure. The delay creator helps the operator to record the reflected signal at certain time-frames. Lastly, the trigger module sends the command to the SPST switch for toggling from non-conductive to conductive condition and vice versa. 

The IAP was elevated from 10 to 24 mmHg by increasing the IAV via the air-filled balloon (instilling around 30 mL of air (at the beginning) for every stepwise 2-mmHg IAP elevation, which decreased exponentially at higher IAP values). At each pressure value, the 10-GHz-transient-radar signal was directed towards the abdominal wall phantom (see Figure 3) and its time-dependent reflection was recorded and further processed by applying advanced signal processing techniques [14]. 

### 2.3. Signal Processing and Statistical Analysis

Data processing in addition to regression analysis were done via MATLAB (MathWorks Inc., Natick, MA, USA). Using the averaging and histogram techniques, the raw signals obtained from the oscilloscope of the TRM set-up were converted to smooth signals. The averaging process was restricted to ten measurements to avoid drift phenomena in the circuit. Density weighted averaging was used in the histogram technique as well [14]. Ultimately, calibration was done by means of regression analysis between the IAP values and the smoothed reflection signals of the abdominal wall phantom. The calibration process resulted in finding a function to calculate the IAP values from the reflection response from the TRM set-up. Subsequently, additional IAP measurements were done (after calibration) in order to evaluate the accuracy and repeatability of the already found function for IAP calculation. The IAP was again elevated from 10 to 24 mmHg (by a blind operator) and the reflection response of the abdominal wall phantom was measured twice at each pressure value. By means of the function that was already obtained by the calibration process, IAP values were calculated and compared with pressure values recorded by the pressure gauge of the air pump.

Values are expressed as mean with standard deviation in case of normal distribution. Pearson regression coefficients were calculated. Considering the research guidelines of the abdominal compartment society [15], Bland and Altman’s analysis is one of the descriptive statistical analysis that should be used in order to evaluate the robustness of a novel IAP measurement method. In Bland and Altman’s analysis, the difference between the two measurement techniques is plotted versus their mean. Additionally, the limits of agreement, calculated as the bias plus or minus twice the standard deviation, should be considered. Consequently, Bland and Altman’s analysis was done to compare the results obtained from the TRM set-up with the real pressure values.

## 3. Results

### 3.1. Monitoring IAP Fluctuations via TRM

The reflection response of the abdominal wall phantom obtained from the oscilloscope is presented in Figure 4. According to the initial results, the best resolution can be obtained at 183.6 nanoseconds (ns). 

By means of the averaging and histogram techniques, the obtained raw signals (Figure 4) were then converted into the smoothed signal, which is illustrated in Figure 5. Considering the 183.6 ns time frame, it is apparent that the voltage difference between the reflection responses at different IAP values was decreased by increasing the IAP, which represents the different IAP elevation stages reported by previous studies [16]. As illustrated in this figure, the IAP elevation from 10 to 24 mmHg resulted in decreasing the reflection response from 8 to 1 mV, approximately.

The measurement resolution is the highest for the IAP values within the range of 10–13 mmHg, which reflects the reshaping phase in IAP elevation. In the stretching (13–19 mmHg) and pressurization (19–24 mmHg) stages, however, the measurement resolution is not as high as in the stretching phase, which is mainly due to the lower expansion of the abdominal wall because of increasing IAV. In fact, thickness reduction and mechanical stretch of the abdominal wall are the main determining parameters in these two stages, which are not as significant as the displacement of the abdominal wall in the reshaping phase. For a better representation, the IAP versus voltage (at 183.6 ns) was also plotted (see Figure 6). The different IAP phases of this study have been determined on the basis of the shape evolution of the abdominal wall phantom.

According to Figure 6, a linear correlation coefficient of −0.97 with a p-value of 0.0001 was observed between the IAP and the reflection response of the abdominal wall phantom.

### 3.2. Absolute IAP Measurement via TRM

In the next step, a quadratic regression analysis (at 183.6 ns) with 95% confidence interval was performed between IAP and voltage values resulting in the equation below:IAP = A × V^2^ + B × V + C,(1)
where A, B, and C are the regression coefficients and V is the voltage value (at 183.6 ns) that can be obtained from the TRM oscilloscope. Table 1 represents the values of A, B, and C in addition to the goodness of fit parameters.

As illustrated in Table 1, the following derived function allows to calculate the IAP with a R^2^ of 0.99:IAP = 0.234 × (V^2^) − 3.949 × (V) + 26.252(2)

Using this quadratic relationship between the IAP and the reflection response of the abdominal wall phantom, 16 additional IAP measurements (2 at each IAP level between 10 and 24 mmHg) via TRM were performed with the same measurement conditions to assess the accuracy and repeatability of the derived function. One more time, the smoothed data was generated using the averaging and histogram techniques. In this part, we only used the voltage values at 183.6 ns as the input variable for Equation (2). Therefore, the absolute value of IAP was calculated based on the reflection response obtained from the TRM set-up. Figure 7 shows the real pressure values according to the pressure gauge of the air pump in addition to the IAP values found according to the quadratic relationship between IAP and the reflection response of the abdominal wall phantom.

Bland and Altman’s analysis was performed according to the reference IAP values obtained from the pressure gauge of the air pump and the IAP values calculated based on the TRM reflection response (see Figure 8). A bias of −0.06 mmHg was seen between the IAP values obtained from the pressure gauge and TRM. The overall mean value ± the standard deviation for the IAP measured with the pressure gauge and TRM was 17.25 ± 4.52 versus 16.48 ± 4.86 mmHg, respectively. Additionally, all the results were within the limits of agreement, defined as the bias plus or minus twice the standard deviation. 

Lastly, according to the results obtained from TRM, accuracy and repeatability of the measurements were calculated as relative error and the standard deviation of mean (SDM), respectively (see Appendix A). Accordingly, the relative error ranged between 0.1 to 6.4% for the first test measurement and 1.3 to 6.1% for the second one, respectively. Additionally, the SDM of the measurements was between 0.18 and 0.4, which represents a relatively high repeatability.

## 4. Discussion

### 4.1. Summary and Interpretation

In this study, a novel non-invasive IAP measurement technique was introduced. The TRM measurement method is a sub-group of the microwave-based techniques that can potentially be used in clinical monitoring. In the previous studies [8,11], frequency-domain microwave reflectometry has been used in order to track the IAP fluctuations in a porcine model, and also in patients undergoing laparoscopic surgery. Based on those results, an almost linear correlation has been found between IAP and the reflection response of the abdominal wall. In general, the main concept behind IAP monitoring via microwave reflectometry is the impact of IAP on the mechanical, geometric, and electromagnetic properties of the abdominal wall and compartment. In general, the IAP elevation can be studied in three different phases known as reshaping, stretching, and pressurization stages [16]. In the reshaping phase, intra-abdominal volume (IAV) increases relatively slowly as a function of IAP, which subsequently results in a relatively significant displacement of the abdominal wall. In the stretching and pressurization phases, however, the IAV expansion due to IAP elevation is not as high as the first stage, which results in high stretch (shear stress) values within the abdominal wall, resulting in steep IAP increases (Figure 9). Depending on the Poisson’s ratio (which is a measure to convert the strain/deformation from one dimension (for instance X) to another dimension (for instance Y)) of the abdominal wall, the generated shear stress decreases the total thickness of the abdominal wall. Ultimately, these fluctuations in the mechanical and geometric characteristics and properties change the dielectric permittivity and permeability of the abdominal wall [11]. 

In this study, we used time-domain microwave reflectometry to monitor IAP changes over time. Using time-domain measurements has the advantage that the displacement of the abdominal wall can be extracted directly from the reflection response, which can potentially result in decomposing the impact of abdominal wall displacement, abdominal wall thickness reduction, shape evolution, etc. Moreover, in this study, we have used a totally contact-free sensor to track IAP, which gives more freedom to healthcare providers to manage the patient under treatment. However, the motion artifact would influence the results and should be filtered out from the reflection response.

In the present in vitro study, IAP was elevated in an abdominal wall phantom by increasing IAV via an air-filled balloon placed behind a water-filled balloon. In former investigations (to simplify the study), we have not used a water layer in the abdominal wall phantom; consequently, a linear relation was found between the amplitude of the reflection signal and the IAP. In this study, however, we added a layer of water to the structure in order to make it more realistic and similar to real case scenarios. After performing further signal processing, the reflection response obtained at different IAP levels was investigated further to understand the correlation between IAP and the voltage value at a certain moment in time (183.6 ns). Quadratic regression analysis was performed in order to find the equation able to calculate the IAP on the basis of the reflection response of the abdominal wall phantom. Subsequently, further random IAP measurements were performed to evaluate the reliability, accuracy, and repeatability of the found equation. By means of the Bland and Altman’s analysis, the robustness of the found model as well as the TRM measurement set-up in IAP monitoring were evaluated. 

The correlation between the reflection response and the IAP values was −0.97 with the *p*-value of 0.0001. The evolution of the reflection response of the abdominal wall phantom versus IAP elevation was in agreement with previous studies [11], since the main principle of IAP monitoring via TRM is based on the geometric, mechanical, and electromagnetic property changes. At the beginning, the reflection response decreased significantly with increasing IAP. However, at higher IAP values, the reflection response reduction was not as high as during the initial stage. This trend can be explained completely on the basis of IAV evolution versus IAP elevation. At the reshaping phase, IAV increases significantly and results in a sharp reduction in the reflection response. In the stretching and pressurization phases, the IAV expansion was not the major determining parameter. In other words, in the stretching and pressurization stages, the reflection response reduction was mainly due to the mechanical stretch and thickness reduction of the abdominal wall, which are not as significant as the impact of IAV elevation in the reshaping phase. 

Using Bland and Altman’s analysis, a bias of −0.06 mmHg was seen between the IAP values obtained from the pressure gauge of the air pump and the TRM set-up. Additionally, all the results were within the limits of agreement, defined as the bias plus or minus twice the standard deviation. Additionally, all the relative error values were within the acceptable limits as defined by WSACS [15].

### 4.2. Future Studies

In future studies, more investigations and quantifications regarding the electromagnetic properties of the abdominal wall phantom should be done in order to correlate the in vitro results with real case scenarios in the ICU. Additionally, due to the inherent properties of the contact-free Transient Radar Methodology, we do not need but could include a distance conserving set-up between the microwave antennas and the abdominal wall. Although it was not a serious challenge to keep the distance between the abdominal wall phantom and the microwave antennas constant in this in vitro study, we can consider both solutions (constant and variable distance) for the clinical applications in the future. For instance, heartbeat and respiration are the main motion artifacts that influence the distance between the abdominal wall and the microwave antennas, and should be considered in the future. Since these artifacts are periodic, they can be removed in the signal processing stage by means of filtering techniques and more advanced signal processing algorithms. Concerning the TRM methodology, we used a frequency of 10 GHz for the measurements. In future studies, an optimal trade-off frequency range will be investigated taking into account the penetration depth and lateral resolution. By increasing the frequency of the waves, we can potentially obtain a better lateral resolution; however, the penetration depth would be less in such a case.

## 5. Conclusions

Although the results of this study showed a promising future for applying TRM in clinical monitoring, it should be pointed out that in this investigation, an advanced abdominal wall phantom was used for IAP monitoring, which is not exactly the same as the human body. In fact, the main focus of this study was on the impact of the IAP elevation on the geometric properties of the abdominal wall, which are more significant compared with the impact of electromagnetic property changes. Additionally, all the reported results were obtained in the lab environment, which is an ideal environment with relatively constant humidity, temperature, etc.; therefore, we could obtain a relatively high accuracy in this study that might not be the case in the ICU. The size of the measurement set-up used in this study also needs to be considered. It is almost impossible to use such a device in clinical monitoring or in critically ill patients. However, a future generation solution would allow a fully integrated transient radar implementation and a more efficient means to manufacture a medical device that can potentially be used in the ICU as well.

## Figures and Tables

**Figure 1 sensors-21-05999-f001:**
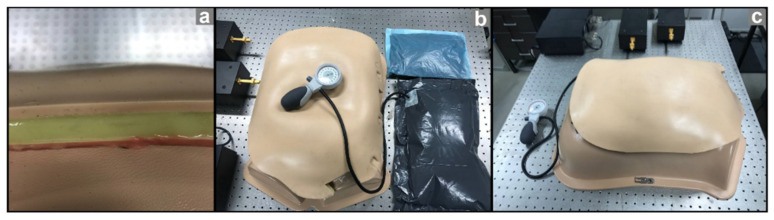
Abdominal wall phantom used in the in vitro study. (**a**) The multi-layer structure of the artificial abdominal wall, representing skin, fat, muscles, fascia, and peritoneum. (**b**) Artificial abdominal compartment in addition to water- and air-filled balloon to simulate different case scenarios in intra-abdominal hypertension. IAV was increased via the air-filled balloon, which subsequently resulted in intra-abdominal pressure (IAP) elevation. (**c**) The final assembly of the abdominal wall phantom studied by transient radar method (TRM).

**Figure 2 sensors-21-05999-f002:**
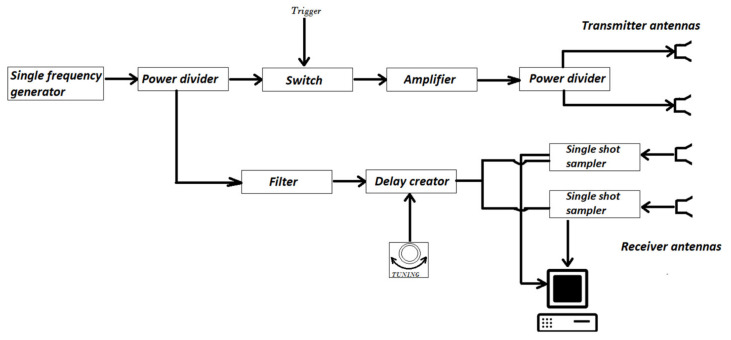
Block diagram explaining different modules of the TRM system.

**Figure 3 sensors-21-05999-f003:**
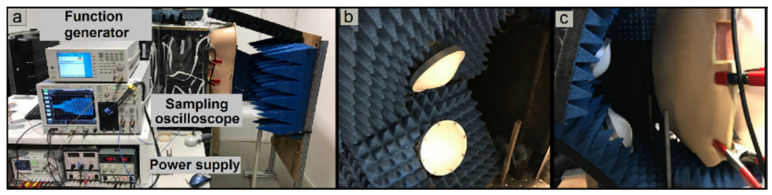
Transient radar method (TRM) set-up used for intra-abdominal pressure (IAP) monitoring. (**a**) TRM set-up in addition to the abdominal wall phantom. In this set-up, the reflection response of the abdominal wall phantom is recorded by means of the sampling oscilloscope. (**b**) Transmitter and receiver antennas in order to radiate electromagnetic waves and receive the reflection signal from the abdominal wall phantom. (**c**) Abdominal wall phantom in front of the transmitter and receiver antennas.

**Figure 4 sensors-21-05999-f004:**
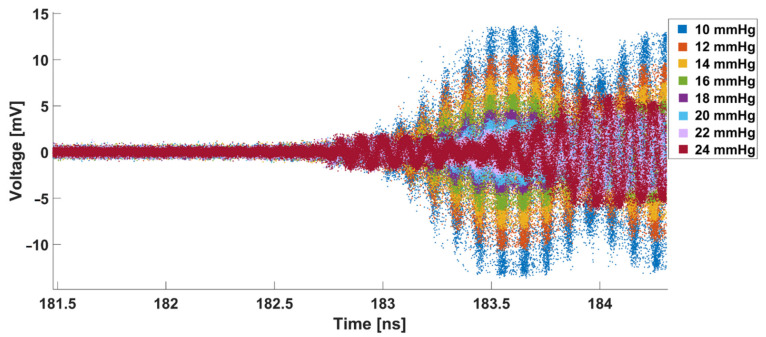
Raw transient signals reflected from the abdominal wall phantom at different IAP values. The reflection response between 181.5 and 182.5 ns is almost zero since there was a distance between the transmitter antenna and the front side of the abdominal wall phantom. Consequently, no reflection happened at this interval. At 182.7, however, the reflection response starts to increase, which means that the first group of reflected waves has been detected by the receiver antenna. Afterwards, more reflected waves from deeper layers of the abdominal wall phantom have been detected and the reflection response has increased progressively.

**Figure 5 sensors-21-05999-f005:**
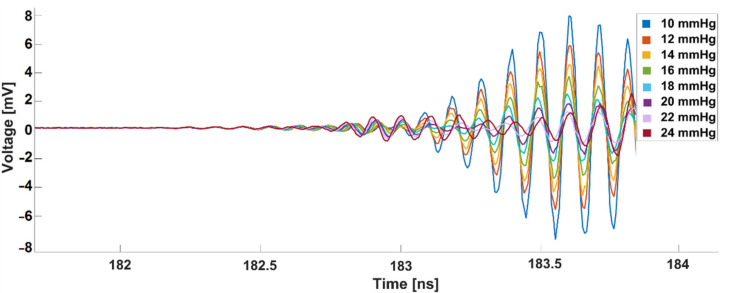
Smoothed reflected signals from the abdominal wall phantom. By means of averaging and histogram techniques, the raw transient signals were converted into the smoothed ones. The highest resolution was found at 183.6 ns.

**Figure 6 sensors-21-05999-f006:**
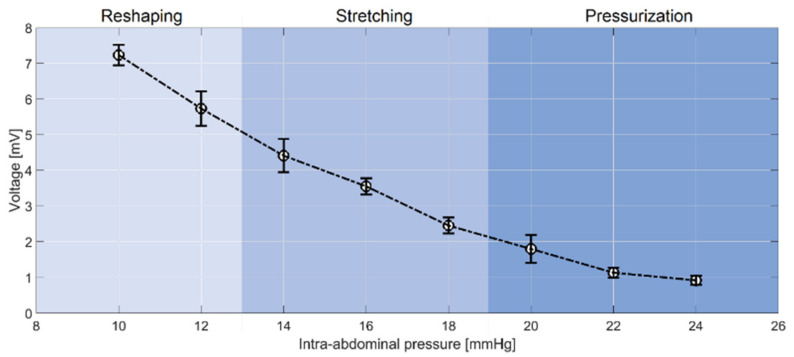
Reflection response of the abdominal wall phantom at different IAP values at 183.6 ns. A quadratic relation was found between IAP and the reflection response of the artificial abdominal wall phantom. Results are represented as mean ± standard deviation.

**Figure 7 sensors-21-05999-f007:**
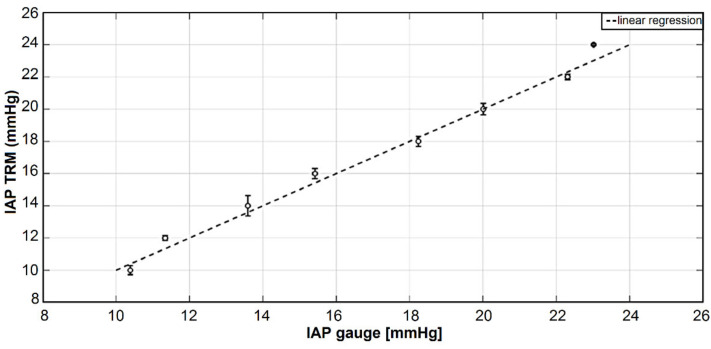
Results of the test measurements obtained after calibrating the TRM system compared with the reference pressure values according to the pressure gauge. As illustrated, the standard deviation values are relatively larger at lower IAP values compared with IAP values higher than 18 mmHg. The main reason for this could be that the displacement variation is higher than the thickness variations in the abdominal wall phantom. Since, at low IAP values, the displacement of the abdominal wall is the main changing parameter, greater standard deviations are obtained at low IAP values.

**Figure 8 sensors-21-05999-f008:**
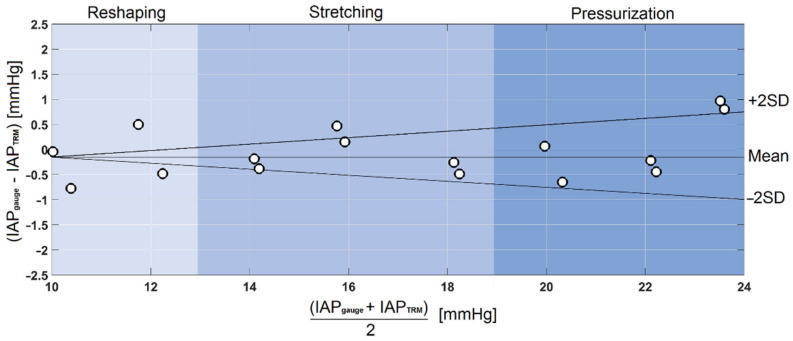
Bland and Altman’s analysis, which shows the mean, bias, and standard deviation of test measurements performed after calibrating the whole system for IAP measurements. A bias of −0.06 mmHg was found between the IAP values obtained from the pressure gauge and TRM.

**Figure 9 sensors-21-05999-f009:**
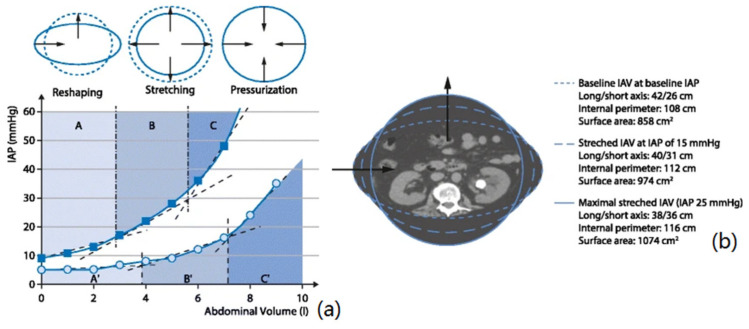
Evolution of intra-abdominal pressure (IAP) versus intra-abdominal volume (IAV) in two different subjects. (**a**) As shown in this figure, intra-abdominal hypertension (IAH) can be studied in three different stages known as reshaping, stretching, and pressurization. At the reshaping phase, slight IAP elevation results in relatively significant IAV increase; however, according to the abdominal compliance, the IAV increase versus IAP elevation is smaller in the stretching and pressurization phases, respectively. (**b**) Patients with an ellipse-shaped internal perimeter have a relatively high abdominal compliance; this is illustrated with the progression of the shape from ellipse (dotted line) at baseline to a sphere (solid line) at very high IAP obtained during laparoscopy. During increase in intra-abdominal volume (IAV) from baseline to stretched and maximal stretched IAV, the difference between the long and short axes of the ellipse decreases, while the internal perimeter and surface area increase. (Adapted from Malbrain et al. [16] with permission).

**Table 1 sensors-21-05999-t001:** Regression coefficients of the quadratic regression analysis between IAP and the voltage values of the reflection response of the abdominal wall phantom. The R^2^ and sum of squares error (SSE) of the regression analysis are shown as well. The SSE shows the total deviation of the response values from the fit to the original values obtained by the pressure gauge.

Regression Coefficients			Goodness of Fit	
**Parameter**	Value	95% CI	Parameter	Value
**A**	0.234	(0.13, 0.33)	SSE	2.304
**B**	−3.949	(−4.82, −3.06)	R2	0.986
**C**	26.520	(25.04, 28.01)		

## Data Availability

Derived data supporting the findings of this study are available from the corresponding author on request.

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
