# Peer review of "Non-Invasive Intra-Abdominal Pressure Measurement by Means of Transient Radar Method: In Vitro Validation of a Novel Radar-Based Sensor"

_sensors, 2021, doi:10.3390/s21185999_

Round 1
Reviewer 1 Report
This paper deals with the development of a novel technique for contactless continuous monitoring of IAP in ICU. The need for such a device is unquestionable and the authors have shown that the applicability is feasible. Well done!
However, two main concerns should be pointed out:
- The distance between the antennas and the abdominal wall should be kelp constant in order to ensure that the measurements are in fact dependent on the abdominal wall's deformation due to IAP.
- The electromagnetic parameters of the phantom need to be studied as well in order to elaborate on the applicability of the technique on tissue.
Overall this is good quality work, which adds valuable knowledge to the field.
Minor comments can be found in the PDF file attached to this review.

Author Response
Dear reviewer,
We appreciate you for reviewing our manuscript and your comments.
The point-by-point reply to your remarks can be found as the attached PDF file.
Yours Sincerely
Authors

Reviewer 2 Report
Dear authors
Congratulation for accomplish this great job. You provided an innovative and non-invasive method to detect and measure intraabdominal pressure which might improve ICU care in the future. I appreciate your effort and results.
I have some questions about this study.
You used an abdominal wall phantom to create the experimental model of IAP. Then used TRM and transfer the signal to calculate the IAP values. It sounds reasonable. But two questions raised.
First, the content of the phantom is homogenous medium , however, the contents in the abdominal cavity are not homogeneous. There will be solid and hallow organs in the human abdomen. I wonder the performance of TRM in heterogeneous contents perform as good as in homogenous one or not ? Please comment it and offer references to support.
Second, following above question, the hallow organs can move in human abdomen. The moving objects influences the reflection of radar wave which might affect the further calculation of IAP value. Please comment this.
Thank you again to have the honor to review this article.
Best wish,
Author Response

(The authors gave the same response as above.)

Round 2
Reviewer 2 Report
Dear authors
Thank you for your revision and the main issues were answered well.
I have no other questions about this article.
Congratulation again for this work.
Best wish,